# Development and Appraisal of a Web-Based Decision Aid for HPV Vaccination for Young Adults and Parents of Children in Israel—A Quasi-Experimental Study

**DOI:** 10.3390/vaccines11061038

**Published:** 2023-05-29

**Authors:** Yulia Gendler

**Affiliations:** The Department of Nursing, School of Health Sciences, Ariel University, Ariel 40700, Israel; yulia.gendler@gmail.com

**Keywords:** HPV vaccination, decision aid, shared decision making, decisional conflict, decision self-efficacy

## Abstract

Background: This study aimed to develop and evaluate the effectiveness of two web-based decision aids designed to help parents of children aged 10–17 years and young adults aged 18–26 years make informed decisions about the HPV vaccine. Methods: The decision aids were developed according to the International Patient Decision Aid Standards (IPDAS) criteria and included information about the vaccine, probabilities of benefits and side effects, personal narratives, and values clarification. The study utilized a quasi-experimental design and included 120 Hebrew-speaking parents and 160 young adults. Participants completed baseline surveys and, two weeks after using the decision aid, completed a follow-up survey. Results: Both parents and young adults experienced a reduction in decisional conflict, an increase in self-efficacy, and greater confidence in the safety and effectiveness of the vaccine. The proportion of participating parents deciding to vaccinate their children against HPV increased from 46% to 75%, and the proportion of participating young adults leaning towards receiving the HPV vaccine increased from 64% to 92%. Conclusions: The study highlights the importance of using decision aids to support informed decision making about vaccination and suggests that web-based decision aids may be a useful tool for supporting Israeli parents and young adults to make HPV vaccination decisions.

## 1. Introduction

Human papillomavirus (HPV) is the most widespread sexually transmitted disease worldwide, with over 14 million individuals estimated to be infected annually and a lifetime risk of infection exceeding 80% [1]. HPV infection is a major public health concern as it can contribute to a range of serious health conditions, such as cervical, vaginal, and vulvar cancers in women, penile cancer in men, as well as anal and oropharyngeal cancers, genital warts, and recurrent respiratory papillomatosis in both men and women [2,3].

Vaccination is an effective primary prevention strategy that aims at reducing the incidence of HPV-related diseases [4]. The World Health Organization (WHO) recommends vaccination of boys and girls aged 9 to 14 years against HPV; moreover, the vaccines are licensed for use for up to 26 or 45 years [5]. Global HPV vaccination rates vary widely among countries and regions. In many high-income countries, HPV vaccination coverage can range from 70% to over 90% among the targeted population. However, in low- and middle-income countries, coverage rates can be significantly lower, often less than 50% [3,6]. Low coverage of the HPV vaccine worldwide can be attributed to several factors. These include a lack of awareness about HPV and the vaccine, limited accessibility to healthcare services, high vaccine costs, vaccine hesitancy driven by safety concerns, cultural and social factors influencing acceptance, and challenges within healthcare systems [7,8].

Currently there are three HPV vaccines that are licensed in Israel, all of them containing virus-like particles against high-risk HPV types: Cervarix© vaccine protects against types 16 and 18, Gardasil© and Gardasil-9© vaccines protect against HPV types 6, 11, 16, and 18, while Gardasil-9© also protects against additional types, including 31, 33, 45, 52, and 58 [9].

In 2013, Israel implemented a program to universally vaccinate eighth-grade girls in middle schools and ninth-grade girls at health bureaus, with the Cervarix© vaccine being added to the government-subsidized basket of drugs and medical treatments [10]. In 2015, the program was expanded to include anti-HPV vaccination (Gardasil©) for men and boys aged between 9 and 26 years. Since 2019, Gardasil-9© vaccine is being provided to both eighth-grade girls and eighth-grade boys at schools. Obtaining parental approval is a prerequisite for administering HPV immunization in schools [9,11]. However, despite the vaccine’s safety, efficacy, availability, and inclusion in the health basket (which allows for free administration), HPV vaccination rates in Israel remain low, with annual rates fluctuating between 52% and 64% [11,12], compared to the high coverage rates (≥95%) for other routine vaccines offered in schools [13]. The reasons for low immunization rates are commonly linked to inadequate information for informed decision making, exposure to vaccine-hesitant misinformation in online platforms, or local structural barriers that impede vaccine uptake [14].

To promote HPV vaccination rates in Israel, various strategies were utilized, including communication campaigns, providing training to healthcare providers about the importance of the vaccine, setting up reminder systems for the vaccination, and educational sessions led by healthcare professionals for parents and for young adults who were eligible for the vaccination [9,13,15]. However, studies have shown that despite the interventions, parental knowledge regarding HPV infection, cervical cancer, and HPV vaccination remained low [12,13]. Some parents expressed concerns about the safety and efficacy of the vaccine, as well as the appropriate age for vaccination. Moreover, attitudes to the HPV vaccine were influenced by the cultural and religious context [13,15]. Therefore, the development by a neutral institution of a decision aid tool that takes into account cultural and religious norms could help improve informed decision making about HPV vaccination in Israel.

Decision aids are tools designed to foster patients’ involvement in health care decision making. Unlike traditional patient information materials, decision aids do not aim to persuade patients to select or consent to a specific course of action or to comply with a particular behavior [16]. Instead, decision aids make the decision explicit, provide balanced information about the available options, assist patients in comprehending the potential benefits and harms of each option, and support patients and clinicians in identifying and implementing the health care option most aligned with the patient’s preferences and values [17]. Previous studies have shown that patients who use decision aids tend to have better knowledge, feel more informed, have a clearer understanding of their values, and report greater involvement in the decision-making process, compared with those who receive traditional care. Properly designed decision aids have the potential to decrease patients’ decisional conflict and enhance their confidence and self-efficacy in decision making [17,18,19]. 

Various decision aids have been created in different countries with the goal of promoting adherence to HPV vaccination. For instance, a decision aid developed in Hong Kong for parents and young women making decisions about HPV vaccination resulted in an increase in participants’ intention to vaccinate, along with decreased decisional conflict, improved knowledge, and enhanced self-efficacy [20]. In Canada, an online decision-aid tool was launched to provide valuable information and address concerns about the HPV vaccine, ultimately reducing barriers to vaccination [14]. 

To the best of our knowledge, no decision aid for HPV vaccination has been specifically developed for the Israeli population, considering their cultural and national characteristics. Therefore, the aim of our study was to develop and evaluate a decision aid with value clarification methods, tailored for parents of children aged 10–17 and young adults aged 18–26. We also aimed to present the development process of the decision aid and an assessment of its acceptability and utility. Additionally, we conducted pre/post testing to analyze the impact of the decision aid on participants’ knowledge of the HPV virus, its associated morbidity, and the vaccine, as well as on their vaccination intention, decisional conflict, and decision self-efficacy. 

## 2. Methods

### 2.1. Setting

We designed two web-based decision aids: one aimed at parents of children aged 10–17 years and the other tailored for young adults aged 18–26. The decision aids development process was guided by the International Patient Decision Aid Standards (IPDAS), which specify five phases of decision aid development [21,22]: (1) planning phase—carried out a needs assessment, formed a workgroup consisting of a gynecologist, a pediatrician specializing in infectious diseases, a community health nurse, a decision-making specialist, as well as potential users of the decision aids, namely parents of children aged 10–17 and young adults, and then, finally, developed a project plan; (2) drafting phase—created first draft of the decision aids; (3) consensus phase—elicited feedback from expert stakeholders; (4) field testing phase—carried out the research, as elaborated upon later; (5) critical appraisal—evaluated the information presented in the decision aids by the study participants and others who were not involved in the development process.

Information for the decision aids was gathered from up-to-date systematic reviews of randomized controlled trials on HPV vaccines, as well as current clinical guidelines and the Israeli Ministry of Health website. Each of the decision aids consisted of five elements: information and comparison of options (whether to take the vaccine or not), probabilities of benefits and side effects, personal narratives from parents of children or from young adults describing their decision to vaccinate or not, values clarification, and knowledge assessment. The information covered various aspects related to the papilloma virus, such as the types of infections, available vaccines, their significance and efficacy, vaccination procedure, why boys should get vaccinated, safety information, potential side effects, associated costs, and any contraindications for receiving it. The values clarification section contained various statements, such as: “My primary goal is to safeguard my child from contracting genital warts and cervical cancer” versus “I’m not concerned about my child acquiring genital warts or cervical cancer”; “My child understands that receiving the vaccine doesn’t equate to approval for unprotected sexual activity” versus “I’m worried that my child might misconstrue the vaccine as a ‘green light’ to engage in unprotected sexual behavior”; “Getting the HPV vaccination goes against my beliefs” versus “Administering the HPV vaccine to my child doesn’t conflict with my beliefs”.

Following the completion of the development process, the web-based decision aids were installed on computers located in the “Medical Decision-Making Research Lab” within the Department of Nursing at Ariel University.

### 2.2. Participants

The participants in the study consisted of Hebrew-speaking parents (both mothers and fathers) with children aged 10–17 years old (boys and girls) who were eligible for the HPV vaccine. Additionally, the study included young adults (both men and women) aged 18–26 years old who were eligible for the HPV vaccine but had not yet received it. The participants were recruited on behalf of Ariel University through a call-out distributed in local community clinics and were offered a reimbursement of ILS 50 for their participation.

Exclusion criteria: young adults who had already been vaccinated and parents of children who had already received the HPV vaccine.

### 2.3. Ethical Considerations 

Approval was obtained from the Institutional Review Board of Ariel University (approval code: AU-HEA-YG-20210610-1). Written informed consent was received before the participants began responding to the questionnaires.

### 2.4. Study Design

The study employed a quasi-experimental design to gauge the effect of utilizing the decision aid on the outcome measures.

### 2.5. Procedure

Between January and April 2023, participants were invited to Ariel University’s “Medical Decision-Making Research Lab.” Initially, they were asked to complete a self-administered questionnaire as a baseline measure. Following the baseline survey, in the second step, all participants were given access to the web-based decision aid on a lab computer. Parents of children utilized the decision aid customized for them, while young adults utilized the one developed specifically for them. In the third step, two weeks later, all participants received a follow-up questionnaire using the Qualtrics^TM^ platform. All recruited participants completed all three steps of the research.

### 2.6. Measures

A study questionnaire consisting of four sections was developed, based on empirical literature. The questionnaire was administered before and after the delivery of the decision aid. 

The first section assessed participants’ knowledge about HPV and HPV vaccines using eighteen true/false statements. These statements covered topics such as the prevalence and routes of HPV transmission, the associated morbidity including genital warts and cervical cancer, the administration of the vaccine, the recommended age for vaccination, and knowledge about the importance of PAP smear screening even after receiving the vaccine.

The second section focused on participants’ attitudes towards HPV vaccination in terms of HPV vaccine hesitancy, perceived vaccine safety and effectiveness, and their intention to get vaccinated/have their children vaccinated. To measure perceived vaccine hesitancy, a single item was presented to the participants, asking them to rate the extent of their hesitancy towards receiving the vaccine or vaccinating their children, using an 11-point Likert scale (0 = no hesitation at all, 10 = very hesitant). Perceived vaccine safety was assessed through a single item that asked participants to rate how safe they perceived the vaccine to be for themselves or their children, using an 11-point Likert scale (0 = not safe at all, 10 = very safe). Participants’ perceived vaccine effectiveness was measured through a single item that asked them to rate how effective they felt the vaccine was against HPV, using an 11-point Likert scale (0 = not effective at all, 10 = very effective). Finally, participants were asked a single yes/no question regarding their plans to receive the HPV vaccine or to have their children vaccinated against HPV.

The third section used the Decisional Conflict Scale (DCS), a validated 16-item scale that assesses participants’ perception that their decisions are informed according to their values. The scoring system ranges from 0 (no decision conflict) to 4 (extremely high decision conflict) and is based on 16 individual items. To calculate the total score, the sum of the individual item scores is divided by 16 to obtain the mean item score. This value is then multiplied by 25, resulting in a total score that ranges from 0 to 100. Higher total scores indicate a greater level of decisional conflict. Total scores lower than 25 are associated with implementing decisions; scores above 37.5 are associated with decision delay or feeling unsure about implementation [23,24]. The original DCS questionnaire demonstrated good reliability, with a Cronbach’s alpha coefficient of over 0.78 [24]. The Hebrew version of the questionnaire exhibited excellent internal consistency, with a value of 0.92. 

The fourth section used the Decision Self-Efficacy Scale (DSE), an 11-item scale that measures participants’ self-confidence in their decision-making abilities. Participants rated each item on a scale of 0 (not at all confident) to 4 (very confident), and the total score was calculated by summing the individual item scores, dividing by 11, and multiplying by 25, giving a possible total score range from 0 to 100. Higher scores indicated greater self-efficacy [25]. The Cronbach’s alpha of the original English version was 0.92 [25], and that of the Hebrew version was 0.87. 

Demographic data, such as age, gender, education level, religious affiliation, population group, marital status, number of children, and awareness of the available vaccines, were collected using the baseline questionnaire only. The primary outcomes were participants’ DCS scores and their intentions towards HPV vaccination, while the secondary outcomes included knowledge about HPV and the HPV vaccine, perceived HPV vaccine hesitancy, safety, and effectiveness, and DSE.

Following the consent of the questionnaire authors, the tools for measuring decisional conflict (DCS) and decision self-efficacy (DSE) were translated into Hebrew by two separate professionals. Another expert who was not involved in the original translation carried out back-translation of the questionnaires into English. Additionally, a validation process was undertaken in which a panel of experts, familiar with the intended construct of the questionnaire, evaluated whether the questionnaire items were satisfactory in measuring the construct and whether they were sufficient to evaluate the domain of interest. 

### 2.7. Data Analysis 

To estimate the required sample size, we conducted an a priori power analysis, following the guidance of the authors of the decisional conflict scale (DCS) that we had designated as our primary outcome measure. The power analysis was based on a power (1-β) of 0.8 and an effect size of 0.3, using a two-tailed test. The calculated minimum sample size required for each study group (parents of children aged 10–17 and young adults aged 18–26) was 82 participants. 

Data for the study were collected from the decision aid platform utilized in the ‘Medical Decision-Making Research Lab’ and the Qualtrics™ software, and the analysis was performed using IBM SPSS version 29. Qualitative data were presented as frequencies and percentages. The Shapiro–Wilk test was used to assess the normality of the dataset. Quantitative variables were presented as means and standard deviations, and a paired-samples t-test was used for analysis. The internal consistencies of the questionnaires were assessed through Cronbach’s alpha coefficient. The effect size was measured using Cohen’s d. Statistical significance was set at *p* < 0.05.

## 3. Results

A total of 120 parents of children aged 10–17 years and 160 young adults aged 18–26 participated in the study. The sociodemographic characteristics of the two samples are presented in Table 1 and Table 2.

The mean age of the sample of parents with children aged 10–17 years was 43.6 ± 4.3 years. Out of the total, 72 (60%) were female, 84 (70%) identified as Jews, and 36 (30%) identified as Muslim-Arabs. Additionally, 58 (48.3%) identified as secular, 12 (10%) as traditional, 32 (26.7%) as religious, and 18 (15%) as Jewish Orthodox. The number of participants who held an academic degree was 43 (35.8%). Most of the participants (96.7%) were married, and the majority (85%) had two or more children. A total of 102 (85%) parents indicated that they had prior knowledge about the HPV vaccine before this study. 

The sample consisted of young adults aged 18–26 years, with a mean age of 22.6 ± 2.8 years. Out of the total, 108 (67.5%) were female, 118 (73.8%) identified as Jews, and 42 (26.2%) identified as Muslim-Arabs. Additionally, 87 (54.5%) identified as secular, 18 (11.3%) as traditional, 44 (27.5%) as religious, and 11 (6.8%) as Jewish Orthodox. Most participants (122 or 76.2%) were single, and 74 (46.3%) reported being in a monogamous relationship. A total of 145 (90.6%) participants from the young adults group indicated that they had prior knowledge about the HPV vaccine. 

Most participants agreed that the decision aid was easy to understand (93.3% of participating parents and 92.5% of young adults indicated so) and of suitable length (86.6% of parents and 72.5% of young adults indicated so), and that the amount of information in the decision aid was optimal (81.7% of parents and 76.3% of young adults indicated so). Most participants agreed (91.7% of parents and 88.8% of young adults) that the decision aid presented balanced information and found it helpful for others who may be making the same decision (see Table 3). 

The results of the knowledge questionnaire indicated a significant gap in understanding about the papillomavirus and its vaccine among both parents and young adults. Particularly low percentages of correct answers were observed for certain statements in the baseline questionnaire, summarized as follows. Only 46% of the parents and 32% of the young adults correctly identified that the papillomavirus could cause serious health problems in men. Additionally, 65% of parents and 54% of young adults did not know that most carriers of the virus do not exhibit visible symptoms. Only 34% of parents and 60% of young adults correctly identified that the papillomavirus cannot cause genital herpes. Only 40% of parents and 32% of young adults knew that the vaccine is not intended for women only, and 58% of parents and 64% of young adults were not aware that the vaccine requires multiple doses. In addition, 56% of parents and 68% of young adults did not know that the vaccine cannot cure an infection that occurred before receiving the vaccine. Finally, only 34% of parents and 48% of young adults knew that vaccination against papillomavirus does not eliminate the need for cervical cancer screening tests. The follow-up questionnaires indicated a significant improvement in the percentage of correct answers after the use of the decision aid, with both groups, parents and young adults, showing 78–100% accuracy in answering each of the 18 statements (see Figure 1).

Following utilization of the decision aid, both participating parents and young adults reported lower vaccine hesitancy (the scores decreased from 6.1 to 4.5 within the parents group after utilizing the decision aid and from 4.9 to 2.6 within the young adults group after using the decision aid, *p* < 0.001) and higher confidence in vaccine safety (the scores increased from 6.9 to 8.4 within the parents group after utilizing the decision aid, *p* < 0.001, and from 7.1 to 8.6 within the young adults group after using the decision aid, *p* = 0.002) and in effectiveness (the scores increased from 6.7 to 7.6 within the parents group before and after utilizing the decision aid, *p* = 0.02, and from 6.4 to 8.2 within the young adults group before and after using the decision aid, *p* < 0.001). Both parents and young adults demonstrated a significant increase in decision self-efficacy from baseline to follow-up, with scores increasing from 60.1 to 81.9 in the parents group and from 55.2 to 78.8 in the young adults group (*p* < 0.001). The scores for decisional conflict significantly declined, with mean scores of 40.6 before utilizing the decision aid reducing to 16.9 afterwards within the parents group, and scores of 39.2 beforehand decreasing to 14.4 afterwards within the young adults group (*p* < 0.001). The proportion of participating parents deciding to vaccinate their children against HPV increased from 46% to 75% (*p* < 0.001), and the proportion of participating young adults leaning towards receiving the HPV vaccine increased from 64% to 92% (*p* < 0.001) (see Table 4). 

## 4. Discussion

The International Patient Decision Aid Standards (IPDAS) collaboration describes patient decision aids as “interventions designed to help people make specific, deliberative choices. They make explicit the decision, providing balanced information on the options and outcomes that are relevant to a patient’s health status, and help patients clarify personal values for features of options” [26]. Decision aids aim to increase the involvement of patients in health decision making. These instrument appraisals include different outcome measures such as confidence, involvement in the decision-making process, role preference, uncertainty, values and support, knowledge, and satisfaction with the decision-making process [27].

This study presented a novel web-based decision aid that assists Israeli parents of children aged 10–17 and young adults aged 18–26, who are eligible for the HPV vaccine, in making informed decisions about vaccination. Notably, this is the first study of its kind in Israel. Our findings indicate that the tool was well-received and useful for both parents and young adults, resulting in increased knowledge about HPV vaccination, greater decision self-efficacy, and reduced decisional conflict. Moreover, more participants were inclined to opt for HPV vaccination after using the decision aid. Participants also felt more reassured about the safety and effectiveness of the vaccine and were less hesitant about getting vaccinated.

HPV poses a substantial health burden globally [28], prompting extensive efforts to create effective vaccines targeting the strains that increase the risk of cervical, penile, rectal, and oropharyngeal cancers, as well as genital warts [29]. However, vaccination rates against HPV remain low, which can be attributed to various factors, including parental concerns and lack of knowledge. As a result, parents hesitate to vaccinate prepubertal girls against sexually transmitted infections due to concerns about safety and efficacy, resulting in a delay of vaccination beyond the recommended starting age of 10 to 12 years [30]. Additionally, there is a lack of awareness that boys should also be vaccinated [31,32]. Our study demonstrated various aspects in which HPV-vaccine-related knowledge was lacking among parents as well as among young adults, such as the fact that carriers of HPV are mostly asymptomatic, the number of doses that need to be taken, and the need to vaccinate both boys and girls. 

Numerous education-based interventions have been implemented to address these knowledge gaps and concerns, hypothesizing that increased awareness will lead to greater compliance with HPV vaccination [30,33]. Nonetheless, previous studies have demonstrated that educational material alone is insufficient for boosting HPV vaccination rates, as other factors beyond knowledge deficiency influence this decision [33,34]. Making a decision about vaccination can be a challenging and uncertain process because of concerns about potential side effects, fear of future regret, the necessary altruistic effort involved, and the numerous sources of information available, all with varying degrees of credibility [34].

The concept of the decision aid appears to encompass a range of tools designed to improve the process of decision making. In the context of vaccination, it involves identifying the decision to be made—whether or not to get vaccinated—and presenting accurate information about the diseases that can be prevented through vaccination. A decision aid also provides personalized information to assist patients in clarifying their underlying values concerning the pros and cons of the decision [17,21]. Its primary objective is to minimize decisional conflict and enable patients to feel that they are making an informed decision that is consistent with their preferences and values [35]. Our research revealed that the use of a decision aid resulted in a significant decrease in decision-making conflict in both study groups. This was attributed to the participants’ increased self-efficacy and improved ability to identify the issues that require consideration and the relevant information that should be taken into account.

A recent systematic review and meta-analysis examined the efficiency of decision aids in vaccination decision making [36]. The analysis assessed three studies [37,38,39] that gauged the impact of decision aids on vaccination intention, and the outcomes suggest that decision aids may heighten vaccination intention. The intervention groups that used decision aids exhibited a moderate increase in the intention to vaccinate as compared to control groups. They also reported less decisional conflict than control groups, and their perceived risk of vaccination and belief in misinformation decreased [36]. Consistent with these studies [37,38,39], we observed positive perception of the vaccine’s effectiveness and safety and an increase in vaccination intention. This tendency was evident both among parents of children aged 10–17 years, who reported a 63% increase in intention to vaccinate their children, and among young adults, who reported a 43% increase in vaccination intention. Prior research conducted by Wegwarth et al. [40] found that an information leaflet with imbalanced information had led to an increase in the intention to receive the HPV vaccine. This was due to a decrease in the number of participants who accurately estimated the incidence rate of cervical carcinoma and an increase in the number of those who overestimated the risk of cervical carcinoma. However, our study revealed that providing balanced information through a decision aid format could enhance the intention to vaccinate, as well as the assimilation of precise information about HPV, its related diseases, and the HPV vaccine.

The World Health Organization (WHO) has recognized vaccine hesitancy as a multi-dimensional phenomenon that poses a global threat to public health, including for vaccines such as the COVID-19 vaccine [41]. While not a primary outcome measure, our study assessed participants’ hesitancy towards the HPV vaccine using a single question. Our results revealed a significant decrease in perceived hesitancy in both study groups after implementing a decision aid intervention. However, to gain a comprehensive understanding of the impact of vaccine-related decision aids on hesitancy, further research is required using validated scales to measure various dimensions of vaccine hesitancy.

### Strengths and Limitations

We employed a systematic method that adhered to evidence-based principles in order to create a first-of-its-kind self-administered decision aid for HPV vaccination for the Israeli population. This involved collecting current information from systematic reviews of randomized controlled trials. The development process of the decision aid was guided by the International Patient Decision Aid Standards (IPDAS). Through the experimental design of the study, we were able to establish a cause-and-effect relationship between the use of the decision aid and measured outcomes, such as intention to vaccinate, decision self-efficacy, and decisional conflict.

The decision aid was evaluated under controlled laboratory conditions. Therefore, it is unclear how many people would actually read or fully engage with unsolicited decision aid materials outside of experimental settings, potentially reducing the effectiveness of the decision aid in real-world conditions. Additionally, our study assessed participants’ decision-making intentions but did not follow up for a prolonged period to observe actual vaccine uptake. Moreover, our evaluation of knowledge levels was limited to short-term measurements, taken two weeks after the intervention, which may have decreased over time. Future studies should also consider the cultural and religious factors that may affect the acceptability of the decision aid, as these aspects were not taken into account in the current results. Finally, our quasi-experimental design lacked a control group, although it was sufficient for our primary goal of assessing the decision aid’s acceptability and utility and understanding its impact on selected outcome measures.

## 5. Conclusions

The objective of the present study, which is the first of its kind in Israel, was to assess the impact of a decision aid on the knowledge of and attitudes and intentions towards human papillomavirus (HPV) vaccination among parents of 10–17-year-old children and young adults aged 18–26 years. Results indicate that the decision aid effectively increased knowledge about HPV and its vaccine, improved attitudes towards vaccination, and increased the intention to vaccinate. Furthermore, the decision aid was found to be easily understandable, of appropriate length, and to present balanced information, rendering it helpful for decision making regarding HPV vaccination.

The study’s results are important for public-health policymakers and healthcare providers as they suggest that decision aids can be an effective tool in increasing HPV vaccine uptake rates. By providing balanced information about the vaccine, decision aids can help individuals make informed decisions about vaccination. This approach can help overcome vaccine hesitancy and contribute to increasing vaccination rates, which is crucial for preventing HPV-related morbidity.

## Figures and Tables

**Figure 1 vaccines-11-01038-f001:**
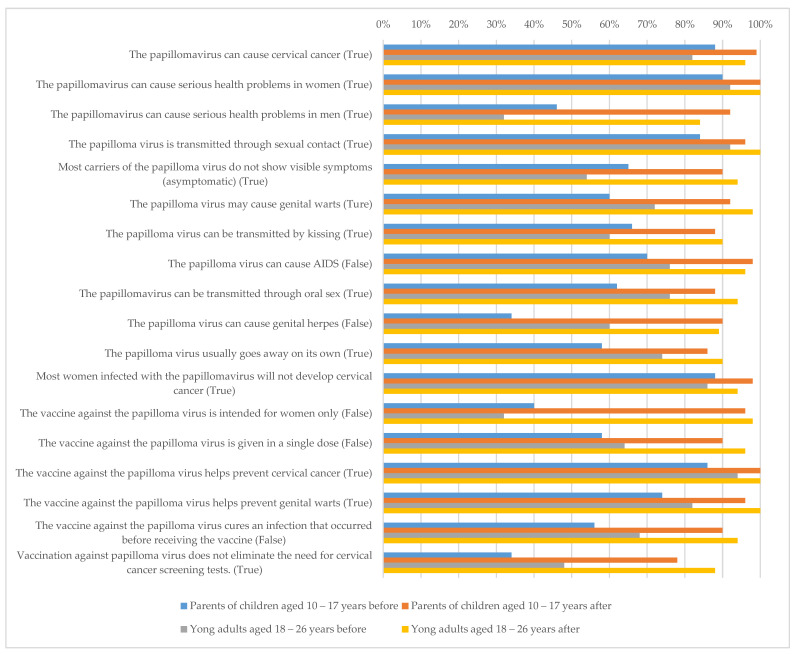
The percentage of correct responses on a knowledge questionnaire related to HPV and the HPV vaccine—before and after utilizing the decision aid.

**Table 1 vaccines-11-01038-t001:** Sociodemographic characteristics of the parents group (*n* = 120).

Variable	*n* (%) or Mean (SD)
Gender	
*Male*	48 (40%)
*Female*	72 (60%)
Age (years)	43.6 (4.3)
Population group	
*Jews*	84 (70%)
*Muslim-Arabs*	36 (30%)
Religious affiliation	
*Secular*	58 (48.3%)
*Traditional*	12 (10%)
*Religious*	32 (26.7%)
*Orthodox*	18 (15%)
Highest education level achieved	
*Primary education*	14 (11.7%)
*Secondary education*	25 (20.8%)
*Diploma*	38 (31.7%)
*Academic degree*	43 (35.8%)
Marital status	
*Married*	116 (96.7%)
*Divorced/separated*	4 (3.3%)
No. of children	
*1*	18 (15%)
*2*	46 (38.3%)
*3*	34 (28.3%)
*4 and above*	22 (18.4%)
Heard of HPV vaccine before	102 (85%)

**Table 2 vaccines-11-01038-t002:** Sociodemographic characteristics of the young adults group (*n* = 160).

Variable	*n* (%) or Mean (SD)
Gender	
*Male*	52 (32.5%)
*Female*	108 (67.5%)
Age (years)	22.6 (2.8)
Population group	
*Jews*	118 (73.8%)
*Muslim-Arabs*	42 (26.2%)
Religious affiliation	
*Secular*	87 (54.4%)
*Traditional*	18 (11.3%)
*Religious*	44 (27.5%)
*Orthodox*	11 (6.8%)
Highest education level achieved	
*Primary education*	20 (12.5%)
*Secondary education*	88 (55%)
*Diploma*	35 (21.9%)
*Academic degree*	17 (10.6%)
Marital status	
*Single*	122 (76.2%)
*Married*	38 (23.8%)
Monogamous relationship	
*Yes*	74 (46.3%)
*No*	86 (53.7%)
Heard of HPV vaccine before	145 (90.6%)

**Table 3 vaccines-11-01038-t003:** Evaluation of information presented in the decision aid.

	Parents of Children Aged 10–17 Years	Young Adults Aged 18–26 Years
	*n* = 120 (%)	*n* = 160 (%)
The decision aid is easy to understand		
*Agree/strongly agree*	112 (93.3%)	148 (92.5%)
*Neutral*	5 (4.2%)	12 (7.5%)
*Disagree/strongly disagree*	3 (2.5%)	0
Length of the decision aid		
*Too long*	8 (6.7%)	38 (23.8%)
*Too short*	8 (6.7%)	6 (3.7%)
*Just right*	104 (86.6%)	116 (72.5%)
Amount of information		
*Too much information*	10 (8.3%)	21 (13.1%)
*Too little information*	12 (10%)	17 (10.6%)
*Just right*	98 (81.7%)	122 (76.3%)
Information presented		
*Encourage to receive the HPV vaccine*	10 (8.3%)	18 (11.2%)
*Discourage to receive the HPV vaccine*	0	0
*Balanced*	110 (91.7)	142 (88.8%)
Others who are making decisions about HPV vaccination may find this decision aid helpful		
*Agree/strongly agree*	114 (95%)	150 (93.8%)
*Neutral*	2 (1.7%)	8 (5%)
*Disagree/strongly disagree*	4 (3.3%)	2 (1.2%)

**Table 4 vaccines-11-01038-t004:** Comparison of the participants’ attitude towards the HPV vaccine, vaccination intentions, decisional conflict, and decision self-efficacy scores before and after utilizing the decision aid (mean (SD)).

	Parents of Children Aged 10–17 Years (*n* = 120)				Young Adults Aged 18–26 Years (*n* = 160)			
	Before	After	*p*	*d ^#^*	Before	After	*p*	*d ^#^*
Vaccine hesitancy	6.1 (3.4)	4.5 (3.1)	<0.001	0.49	4.9 (3.2)	2.6 (1.9)	<0.001	0.87
Vaccine safety	6.9 (3.1)	8.4 (2.2)	<0.001	0.56	7.1 (2.5)	8.6 (1.8)	0.002	0.69
Vaccine effectiveness	6.7 (2.5)	7.6 (2.4)	0.02	0.36	6.4 (3.1)	8.2 (2.8)	<0.001	0.61
Decisional conflict	40.6 (18.8)	16.9 (11.2)	<0.001	1.52	39.2(13.6)	14.4 (6.9)	<0.001	2.29
Decision self-efficacy	60.1 (14.7)	81.9 (16.5)	<0.001	1.41	55.2 (18.4)	78.8 (16.1)	<0.001	1.37
Vaccination intentions	46%	75%	<0.001		64%	92%	<0.001	

*d ^#^*: Cohen’s d effect size.

## Data Availability

The data forming the basis of the findings of this study are available on request from the corresponding author.

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
