# Peer review of "Development and Appraisal of a Web-Based Decision Aid for HPV Vaccination for Young Adults and Parents of Children in Israel—A Quasi-Experimental Study"

_vaccines, 2023, doi:10.3390/vaccines11061038_

Round 1

Reviewer 1 Report

The article is well written and organized. I can give some comments in my field as a statistician:

1. The authors should report the real data along the paper. 

2. Non-parametric plots must be sketched such as quantile-quantile, histogram, kernel density, among others for data to discuss the behavior of data.

3. The normality and homogeneity properties should be tested vis Parametric or non-parametric tests. The authors can use SPSS program. 

4. Abbreviation and statistical tools Sections should be added.

5. What testing instrument did the authors use in the analysis to compare between two groups?.

6. What about the results when using the 0.01 level of significance?.

7. Did the authors calculate the dispersion between groups?. Discuss

No need.

Author Response

Thank you for your time and efforts in reviewing our manuscript. We found the comments and suggestions offered by you to be highly useful.  Accordingly, we incorporated the appropriate changes within the current version of the manuscript. Point-by-point responses to each of the comments are provided below.

  1. The real data is now presented throughout the entire manuscript.
  2. No non-parametric data was subjected to comparison in the study. Only the parameters with a normal distribution, confirmed by the Shapiro-Wilk normality test were compared. Therefore, a paired samples t-test was utilized.
  3. Shapiro-Wilk test was used to assess the normality of the dataset (added to the data analysis section).
  4. Abbreviation section was added, and the statistical tools are described with the "Data analysis section".
  5. The two groups cannot be directly compared as they consist of different populations that utilized distinct decision aids specifically tailored to each group as an intervention. Therefore, no statistical analysis was conducted to compare the two groups.
  6. The t-test results indicated that all the parameters analyzed yielded a p-value of less than 0.01.
  7. Due to the separate and non-comparable nature of the groups, we did not calculate the dispersion between them. Therefore, the groups were presented separately.

Reviewer 2 Report

well planned study and well presented

this would be useful to replicate in other settings

Author Response

Thank you for your time and efforts in reviewing our manuscript. Thank you for your suggestion. We are planning to explore this study in a real-world setting and develop additional decision aids for various medical domains.

Reviewer 3 Report

Thank you for this interesting study. It is important as HPV vaccines are an excellent tool to decrease Cervical Cancer incidence. However, the vaccine coverage is low (or below the target level) in many countries worldwide (if available at all). So, the manuscript has value for publication. However, some improvements should be done before. Please see below my comments:

1. The title is too long. Suggest to crete a more consice variant. Suggested - "Development and Appraisal of Web-Based Decision Aid for HPV Vaccination for Parents in Israel – a Quasi-experimental Study". No need to go into details in the title as Patents are anyway parents (of Children and Young Adults).

2. The abstract represents the study well. Please avoid using abbreviations in the abstract; otherwise, explain them in the text.  IPDAS?

3. The introduction provides a clear rationale for the study, however, appears too long. However, since there are no limitation rules from the journal side, I would leave it up to the authors to decide the length of this part. Moreover, I believe the worldwide HPV vaccine coverage should be mentioned before writing about coverage in Israel. 

4. Also I would suggest mentioning factors leading to low coverage and negative HPV vaccine attituded in general before reporting those factors for the Israelian population. Possible recent resources are - doi: 10.1177/17455057231172355; doi: 10.3390/vaccines10050824; doi: 10.1371/journal.pone.0261203. eCollection 2021.

5. The methods part is very detailed and includes all required parts.

6. The results section is interesting and supported by informative tables and figures.

7. In the discussion section please mention the study's strength alongside the limitations. 

Author Response

Thank you for your time and efforts in reviewing our manuscript. We found the comments and suggestions offered by you to be highly useful.  Accordingly, we incorporated the appropriate changes within the current version of the manuscript. Point-by-point responses to each of the comments are provided below.

  1. The study focuses on two distinct groups: parents of children in the first group, and independent young adults aged 18-26 in the second group, who make decisions for themselves separate from their parents. It was crucial to highlight both study groups in the title. The participating groups in the title were reversed to emphasize their separate and non-comparable nature.
  2. The abstract now provides an explanation of the abbreviations used for IPDAS.
  3. Thank you for that comment. We believe that all the information presented in the introduction section is highly relevant to the readers of this paper, and we have now included the information about global coverage of the HPV vaccine.
  4. We elaborated on the reasons for low vaccination rates and incorporated one of the suggested papers.
  5. Thank you for your support for this section.
  6. Thank you for your support for this section.
  7. The strengths of the study were added to the discussion section.

Round 2

Reviewer 1 Report

All comments have been done.